# Dual-phase nano-glass-hydrides overcome the strength-ductility trade-off and magnetocaloric bottlenecks of rare earth based amorphous alloys

Liliang Shao[1,2,3], Qiang Luo[1,5] ✉, Mingjie Zhang[1], Lin Xue[4], Jingxian Cui[1], Qianzi Yang[1], Haibo Ke[2,5] ✉, Yao Zhang[1], Baolong Shen[1,5] ✉ & Weihua Wang[2,3]

Metal-hydrogen systems have attracted intense interest for diverse energy-related applications. However, metals usually reduce their ductility after hydrogenation. Here, we show that hydrogen can take the form of nano-sized ordered hydrides (NOH) homogeneously dispersed in a stable glassy shell, leading to remarkable enhancement in both strength and ductility. The yield strength is enhanced by 44% and the plastic strain is substantially improved from almost zero to over 70%, which is attributed to the created NOH and their interplay with the glassy shell. Moreover, the hydride-glass composite GdCoAlH possesses a giant magnetic entropy change ($-\Delta S_M$) of 18.7 J kg$^{-1}$K$^{-1}$ under a field change of 5 T, which is 105.5% larger than the hydrogen-free sample and is the largest value among amorphous alloys and related composites. The prominent $\Delta S_M$-ductility combination overcomes the bottlenecks of amorphous alloys as magnetic refrigerants. These results provide a promising strategy for property breakthrough of structural-functional alloys.

The interaction between hydrogen and metals is a topic of great significance in materials science. With increasing attention on hydrogen, an energy carrier of the future, hydrogen-metal systems have been intensively explored for various energy-related applications, such as hydrogen storage and sensing, rechargeable batteries, catalysis, refrigeration, and so on[1–3]. On the other hand, many metals and alloys suffer from hydrogen embrittlement[4,5]. Hydrogen embrittlement becomes a critical issue in many industrial applications of various alloys, including high-strength steels, aluminum alloys, titanium alloys, and magnesium alloys, etc., and thus has drawn much research attention over 100 years[4–6]. Although several mechanisms were proposed for the crystalline metals/alloys-hydrogen systems, the nature of hydrogen embrittlement is not well understood except for a few Zr, Ti

metals/alloys (where the hydride formation leads to embrittlement)[6]. It was found that amorphous alloys (AMAs) may show better resistance to hydrogen embrittlement than crystalline alloys, but they become rather brittle above a critical hydrogen concentration as well[7–9]. Thus, it is widely and urgently needed to develop advanced hydrogen-metal systems combining excellent functional and mechanical performances.

Different from conventional metals and alloys, AMAs have a unique disordered atomic structure with no grain boundaries and dislocation. This endues AMAs with high strength, improved wear resistance, excellent soft magnetic and anti-corrosion properties, and so on[10–12]. Nevertheless, the AMAs have several inherent shortcomings impeding their development and wider applications. Besides the

[1]School of Materials Science and Engineering, Jiangsu Key Laboratory of Advanced Metallic Materials, Southeast University, Nanjing 211189, China. [2]Songshan Lake Materials Laboratory, Dongguan 523808, China. [3]Institute of Physics, Chinese Academy of Sciences, Beijing 100190, China. [4]College of Mechanics and Materials, Hohai University, Nanjing 211100, China. [5]These authors jointly supervised this work: Qiang Luo, Haibo Ke, Baolong Shen.
✉e-mail: q.luo@seu.edu.cn; kehaibo@sslab.org.cn; blshen@seu.edu.cn

restricted sample size imposed by the critical cooling rate for glass formation and limited working temperature below the glass transition temperature, their poor plasticity at ambient temperature and propensity for catastrophic failure originating from severe plastic-strain localization in shear bands (SBs) are critical drawbacks[11], largely undercutting their structural and functional utilization. Moreover, some functional properties cannot be effectively enhanced and even may be degraded through the amorphization of alloys, such as magnetocaloric performances. Although the AMAs take the advantages of magnetic softness, large resistivity leading to smaller eddy current loss, and broad magnetic entropy change ($\Delta S_M$) peak as promising magnetic refrigerant[13,14], they suffer from the drawback of low-to-medium $\Delta S_M$ owing to the amorphous nature with second-order magnetic transition. Among various AMA systems, the Gd-based AMAs showing much larger maximum $\Delta S_M$ ($-\Delta S_M^{pk}$) than the Pd-, Fe-, Ni- and Co-based AMAs[13,15–18] and much better magnetic softness than other rare earth-based AMAs (like Dy-, Ho-, Tb-based)[19,20], are the most investigated amorphous systems. Nevertheless, a recent summary of $-\Delta S_M^{pk}$ values under a field change of 5 T for the Gd-based AMAs suggests that there may exist a limiting value of $-\Delta S_M^{pk} \sim 11$ J kg$^{-1}$ K$^{-1}$ due to the intrinsic disordered structure lacking the strong structurally-magnetically coupled transition[21]. The improvement of $-\Delta S_M^{pk}$ for Gd-based AMAs by composition design, annealing, irradiation, and cryogenic thermal cycling is rather limited[22–25]. Moreover, the brittleness of all magnetic AMAs[26,27] becomes another critical bottleneck for their practical application in magnetic cooling apparatus.

Here, to address the hydrogen embrittlement of metal-hydrogen systems and the strength-ductility trade-off and magnetocaloric bottlenecks in AMAs, we create a Gd-based supra-nanometer-sized dual-phase glass-hydride (SNDP-GH) material, comprising GdH$_2$ grains of around 3.6 nm in diameter, uniformly embedded in the remaining amorphous shell with thickness around 1–4 nm. Strikingly, different from hydrogen embrittlement generally occurs in many metals and alloys, a hydrogen-induced brittle-to-ductile transition is observed even the hydrogen concentration approaches its saturation (~1.5 wt.%). The yield strength is enhanced from 2.5 to 3.6 GPa (by 44%), and plastic strain increases substantially from 0 to (>)70%, breaking the strength-ductility trade-off in AMAs and related composites. Moreover, due to the formation of high-density nanosized ordered hydrides (NOH) in the amorphous matrix, the $-\Delta S_M^{pk}$ of the alloy increases considerably from the initial 9.1 J kg$^{-1}$ K$^{-1}$ (hydrogen-free AMA) to a giant value of 18.7 J kg$^{-1}$ K$^{-1}$ under a field change of 5 T. The combination of giant $\Delta S_M$ and superior mechanical performance overcomes the long-standing performance bottlenecks of AMAs as promising magnetic refrigerants.

## Results

### Hydrogenation and microstructure analysis

The Gd$_{55}$Co$_{17.5}$Al$_{27.5}$ alloy was chosen for the present study since it possesses a large magnetocaloric effect (MCE) and excellent glass-forming ability with a critical diameter of 7 mm for glassy rod formation[28]. In the present study, Gd$_{55}$Co$_{17.5}$Al$_{27.5}$ powders were prepared by high-pressure argon gas atomization method. Figure 1a shows the scanning electron microscope (SEM) image of the Gd$_{55}$Co$_{17.5}$Al$_{27.5}$ powders with a diameter range of 25–30 μm. Most of the particles are spherical. The transmission electron microscopy (TEM) images (Fig. 1b, d) show a typical amorphous structure of the powders without any nanocrystals, which is also verified from the selected area electron diffraction (SAED) patterns (Fig. 1c). Figure 1e

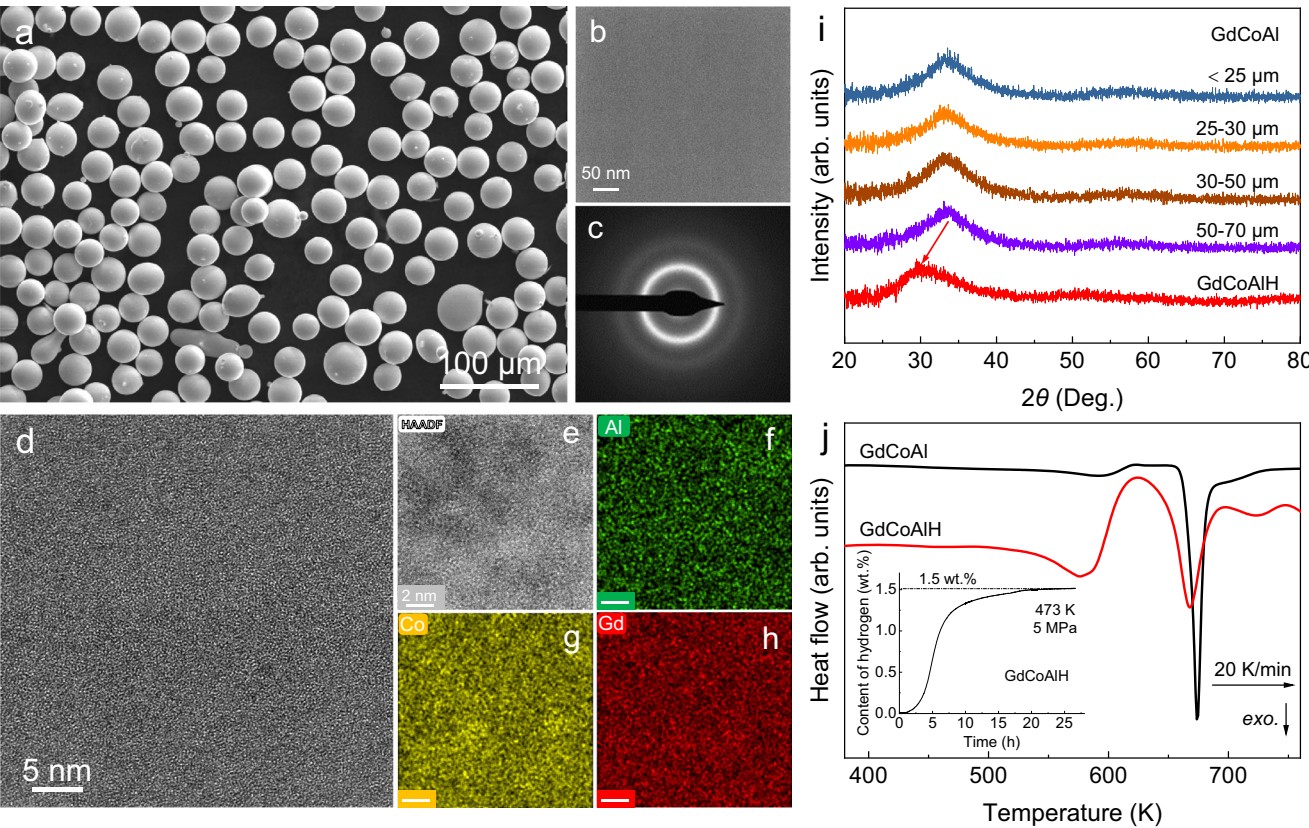

**Fig. 1 | Structural and thermal characterizations of the GdCoAl amorphous powders and isothermal hydrogenation. a** SEM image of the amorphous powders with a diameter of 25–30 μm. **b** TEM image and **c** SAED pattern of the powder. **d** HRTEM image of the powder. **e**–**h** High-resolution HAADF image and corresponding EDS mappings of Al, Co, and Gd elements. **i** XRD patterns of the GdCoAl powders with different diameters less than 70 μm and the GdCoAlH powder with the saturated absorption of hydrogenation. **j** DSC curves of the GdCoAl and GdCoAlH powders at the heating rate of 20 K/min. The inset displays the curve of hydrogenation dynamics.

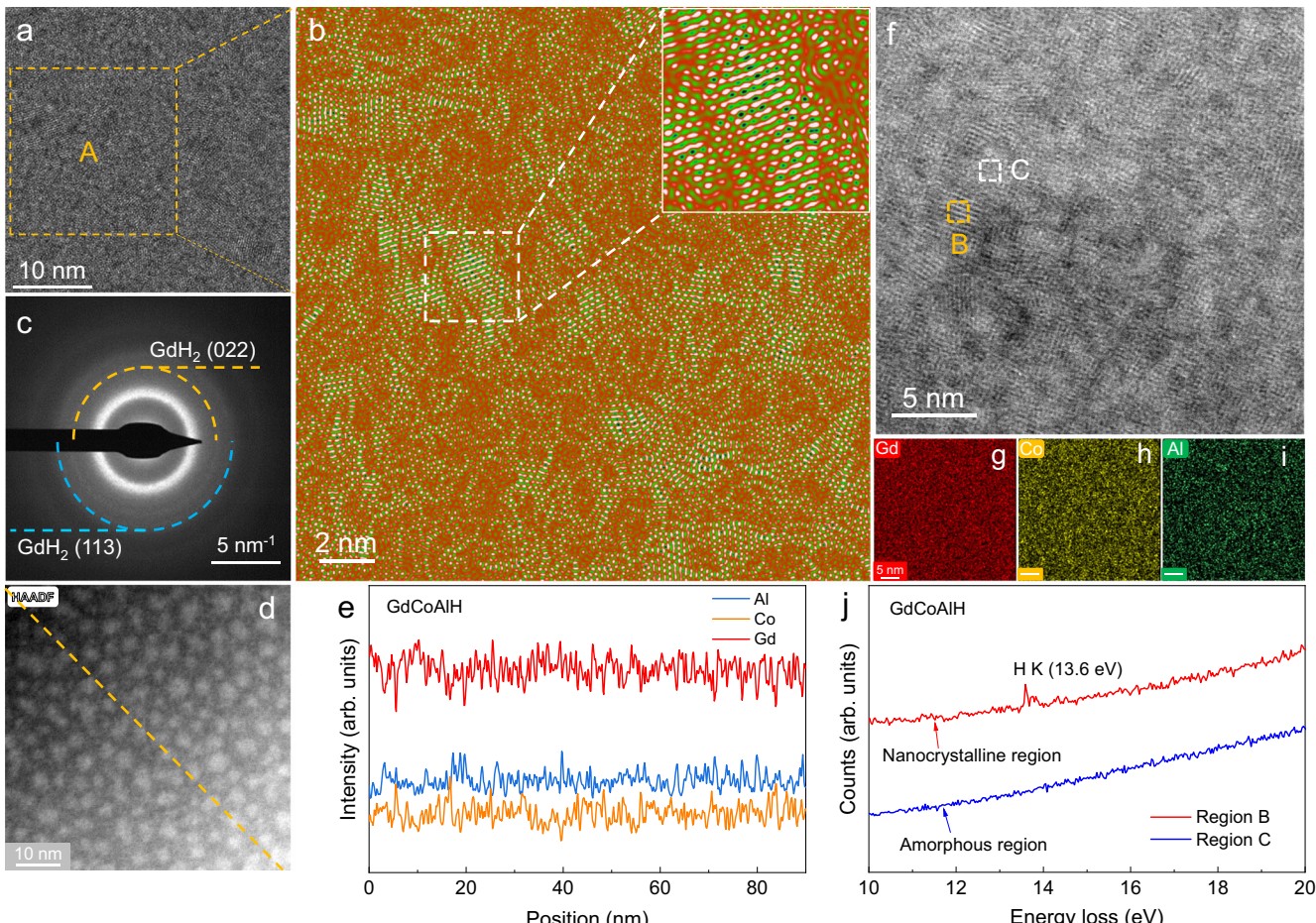

**Fig. 2 | Nanoscale structure and elemental analyses of the GdCoAlH powder. a** HRTEM image and **b** the further magnification of region A (after inverse Fourier transform) in **a**, showing the local ordering structures induced by hydrogenation. **c** Corresponding SAED pattern. **d**, **e** HAADF image and element fluctuation along the diagonal. **f**–**i** HRSTEM image and corresponding EDS mappings of Gd, Co, and Al elements. **j** EELS curves of the regions B (local ordering structure) and C (adjacent amorphous structure) in (**f**).

presents the high angle annular dark-field scanning transmission electron microscopy (HAADF-STEM) image of the powder, in which relatively dark and bright domains can be observed indicating the structural heterogeneity at nanometer scale. Energy-dispersive spectrometry (EDS) mappings of the same region are shown in Fig. 1f–h, which show homogenous distributions of the Gd, Al, and Co elements and do not correlate apparently with heterogeneity seen from the HAADF-STEM image. Thus, the contrast variation in the STEM image mainly arises from the density fluctuation, which is an intrinsic feature of AMAs.

AMAs have a disordered structure and a wide spectrum of atomic-packing heterogeneities, which provide an infinity of interstitial sites for hydrogen occupation[8]. For the present $Gd_{55}Co_{17.5}Al_{27.5}$ AMA containing Gd element with a strong affinity for hydrogen, a high hydrogen concentration and formation of hydride in the alloy can be expected after hydrogenation. Figure 1i shows the X-ray diffraction (XRD) patterns of the as-prepared $Gd_{55}Co_{17.5}Al_{27.5}$ and hydrogenated powders (GdCoAl and GdCoAlH for short, respectively). The broad diffraction maxima of the pattern shifts to lower $2\theta$ after hydrogenation, indicating a volume expansion as observed in other AMAs[29,30]. The differential scanning calorimetry (DSC) curves of the GdCoAl and GdCoAlH powders are presented in Fig. 1j. Distinct endothermal events corresponding to glass transition and exothermal event resulting from crystallization are observed for the as-prepared powders. And the onset temperatures for glass transition and crystallization are determined to be 600 and 670 K, respectively. A large supercooled-liquid

region of 70 K, indicating its excellent glass-forming ability. After hydrogenation, the crystallization temperature decreases slightly to around 667 K. Due to the overlap between the glass transition and hydrogen desorption (starting around 600 K), the onset temperature of the glass transition cannot be determined accurately. The existence of a supercooled-liquid region and crystallization peak after hydrogenation indicates that there remains a significant amount of amorphous phase.

To explore the interaction between hydrogen and Gd-based AMA in more details, the atomic structure of the hydrogenated alloy was further investigated by special aberration-corrected high-resolution TEM (HRTEM) and HAADF-STEM. The HRTEM image indicates that the sample possesses a unique structure with some structurally ordered regions with several nanometers in size embedded in the amorphous matrix (Fig. 2a). This can be further seen from the inverse Fourier transform (IFT) image (Fig. 2b) of region A in Fig. 2a, in which lattice stripes (white area) were observed. Moreover, the SAED pattern reveals the presence of the $GdH_2$ phase (Fig. 2c), which implies that the ordered region may be the $GdH_2$ clusters. The STEM image indicates more clearly that many nanosized (2–5 nm) ordered hydrides (NOH) are embedded homogeneously in the amorphous matrix (Fig. 2d). The amorphous matrix serves as a shell with a thickness of 1–4 nm, which wraps up the NOH structure. And the concentration profiles of Gd, Al, and Co elements (Fig. 2e) along the diagonal line of the STEM image in Fig. 2d show a random fluctuation without any correlation with the contrast variation observed in Fig. 2d. This means that there is no

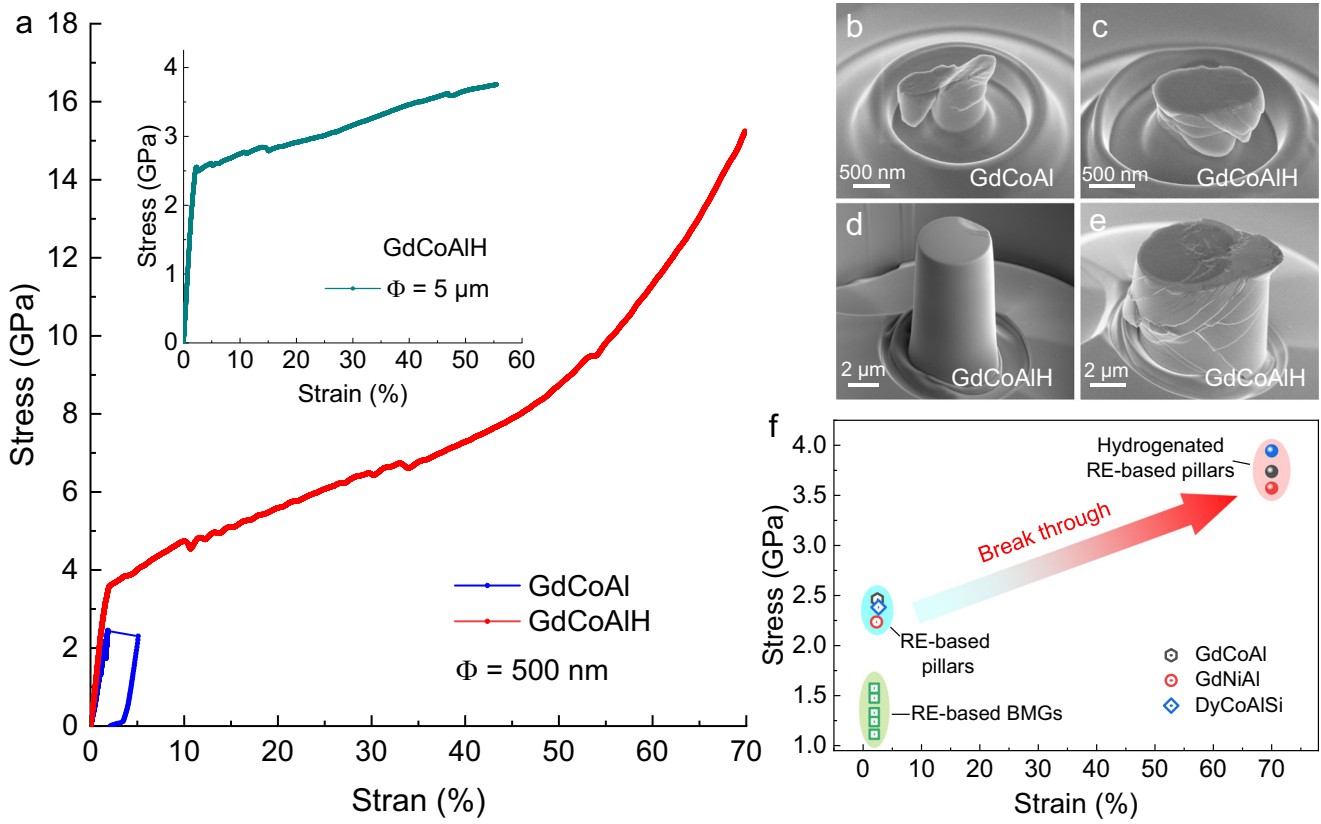

**Fig. 3 | Mechanical properties of the GdCoAl and GdCoAlH micropillars.**
**a** Engineering stress-strain curves of the pillars with a diameter (Φ) of 500 nm and height of 1 μm. The inset shows the stress-strain curves of the GdCoAlH pillar with a diameter of 5 μm and height of 10 μm. **b**, **c** Morphology images after compression of GdCoAl and GdCoAlH pillars with a diameter of 500 nm. **d**, **e** Morphology images of the GdCoAlH pillar with a diameter of 5 μm before and after compression. **f** Summary of the stress versus strain for the typical rare earth-based amorphous micropillars and BMGs, the data of this study are also added.

apparent concentration difference of Gd, Al, and Co between the NOH and amorphous matrix. High-resolution STEM (HRSTEM) has been carried out to further characterize NOH regions (Fig. 2f). The NOH regions can be observed clearly, which comprise several ordered clusters. Element mappings in Fig. 2g–i present a homogenous distribution, which agrees with the results shown in Fig. 2e. This means that hydrogenation does not induce aggregation of elements, and the hydrogen atoms just occupy the interstitial sites and most of them interact with their neighboring Gd atoms to form NOH. Electron energy loss spectroscopy (EELS) was further used to characterize the structurally ordered regions. Figure 2j shows the EELS spectra for the typical ordered region B and disordered region C. A peak at 13.6 eV from the hydrogen K edge is clearly observed for region B (NOH) in Fig. 2f, which is absent for region C. In a word, the Gd-based SNDP-GH material comprises the randomly distributed $GdH_2$ clusters of around 3.6 nm in diameter and the remaining amorphous shell with a thickness of 1–4 nm. The similar structures were also constructed in the GdNiAl and DyCoAlSi AMAs via hydrogenation (Supplementary Fig. 1).

## Mechanical properties

The brittleness of AMAs arising from a lack of crystalline defects is one of the critical problems hindering their wider practical applications as structural and/or functional materials, including magnetic refrigerant. It is well-known that many crystalline alloys are subjected to hydrogen embrittlement[4,5]. Hence, it is of interest to explore the mechanical performance of this hydrogenated alloy with a dual-phase structure. Since powders were investigated in the present study, we carried micro-compression tests on the micropillars fabricated from the powders. Figure 3a compares the compressive engineering stress-strain curves of the pillars with a diameter of 500 nm and a height of

1 μm for the as-prepared and hydrogenated alloys. The GdCoAl sample has a yield strength of 2.5 GPa without any plasticity, indicating its intrinsic brittleness. In contrast, the hydrogenated sample shows a yield strength of 3.6 GPa and a plastic strain of over 70%, reversing the hydrogen embrittlement in many metals and alloys. Larger size sample with 5 μm in diameter was also investigated and the result is shown in the inset of Fig. 3a. A large strain of over 55% (without fracturing) is still observed, illustrating that such a brittle-to-ductile transition does not arise from the size effect. Figure 3b,c shows the morphology images after compression of the GdCoAl and GdCoAlH pillars with a diameter of 500 nm. The GdCoAl pillar shows the features of brittle fracture. In contrast, the GdCoAlH pillar was compressed into a flake without fracturing and obvious SBs are observed. The multiple SBs can be seen more clearly for the GdCoAlH pillar with 5 μm in diameter (Fig. 3e). Furthermore, we investigated different systems (GdNiAl and DyCoAlSi), and similar brittle-to-ductile transition induced by hydrogenation is observed (Supplementary Fig. 2). Figure 3f shows the excellent mechanical performances of the several hydrogenated rare earth (RE)-based AMAs compared with their hydrogen-free counterparts and other typical RE-based bulk metallic glasses (BMGs). It can be seen clearly that the SNDP-GH nanostructure enhances greatly both the strength and plasticity, overcoming the strength-ductility trade-off in AMAs. Meanwhile, these results imply that creating an SNDP structure could be an effective strategy to overcome the hydrogen embrittlement of many metal-hydrogen systems.

## Magnetic transition and magnetocaloric properties

Next, we investigated the effect of hydrogenation on the magnetic transition and magnetocaloric properties. Figure 4a presents the temperature dependence of the zero-field-cooling (ZFC) and

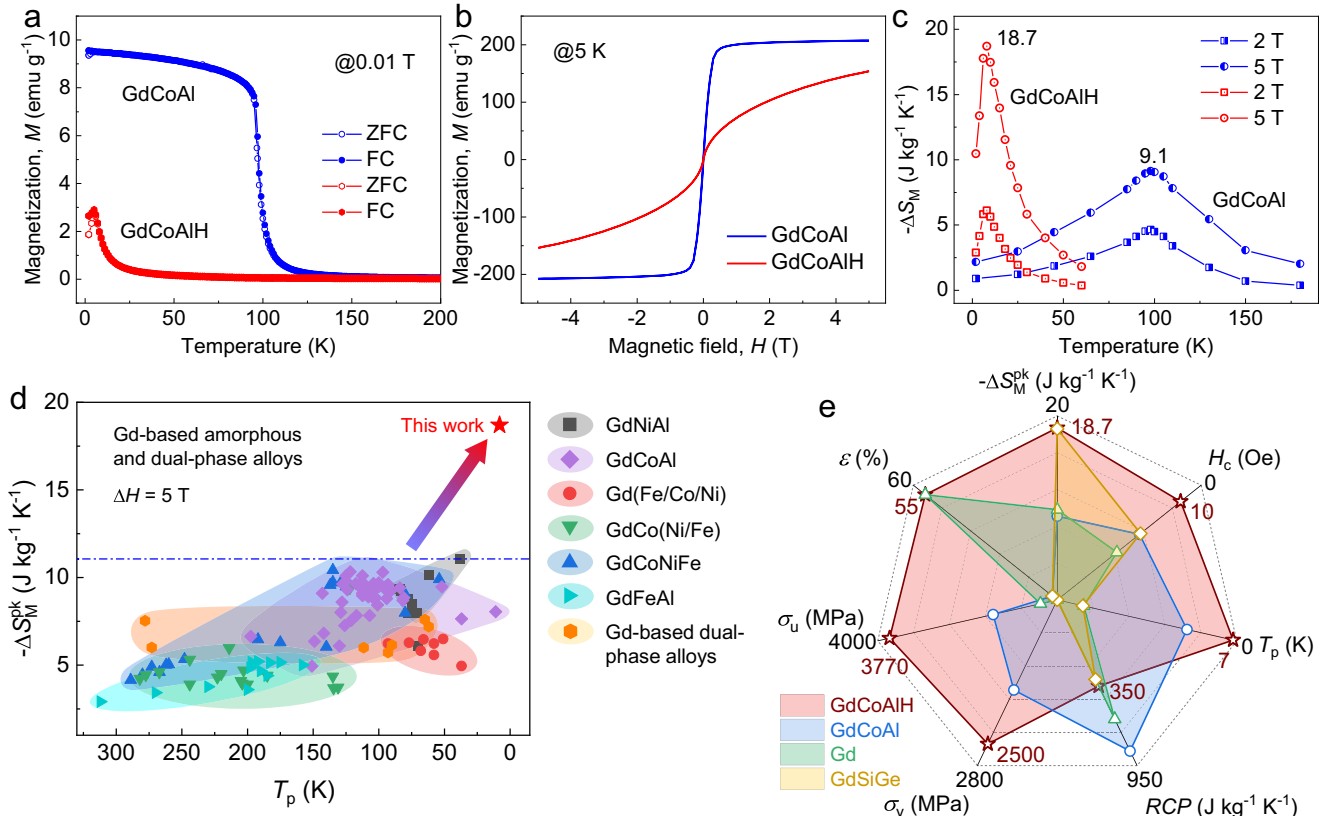

**Fig. 4 | Magnetic transition and magnetocaloric effect of the GdCoAl and GdCoAlH powders. a** Temperature dependence of magnetization ($M$). **b** Magnetic hysteresis loops measured at 5 K. **c** Temperature dependence of magnetic entropy change ($\Delta S_M$) under the maximum applied field of 2 and 5 T. **d** Summary of maximum magnetic entropy change ($-\Delta S_M^{pk}$) (magnetic field change $\Delta H = 5$ T) versus the peak temperature ($T_p$) for the typical Gd-based amorphous and dual-phase alloys, as well as the value of GdCoAlH. **e** Comparison of the mechanical (ultimate strength $\sigma_u$, yield strength $\sigma_y$, and strain $\varepsilon$) and magnetic properties ($-\Delta S_M^{PK}$, $T_p$, coercivity $H_c$, and relative cooling power $RCP$) of the GdCoAl and GdCoAlH alloys and representative crystalline Gd and GdSiGe alloys.

field-cooling (FC) magnetization of the GdCoAl and GdCoAlH alloys. The same temperature dependence of the ZFC and FC curves is observed for GdCoAl, indicating a typical ferromagnetic transition. In contrast, for the GdCoAlH alloy, both the ZFC and FC magnetization curves show a peak around the Néel temperature $T_N$, below which bifurcation appears between the FC and ZFC branches. From the Curie-Weiss fit of the magnetic susceptibility far above the magnetic ordering temperature (Supplementary Fig. 4a), the paramagnetic Curie temperatures are determined to be 107 K and −27 K for the GdCoAl and GdCoAlH alloys, respectively. This verifies that ferromagnetic and antiferromagnetic exchange interactions dominate the magnetic structure in GdCoAl and GdCoAlH, respectively. The AC susceptibility $\chi$ (real part) curves under different frequencies from 13 to 9673 Hz are almost the same (Supplementary Fig. 4b), indicating further the antiferromagnetic transition nature, which is different from the obvious change of the peak position and intensity with increasing frequency in the Ho- and Dy-based AMAs showing spin-glass-like behavior[19] (Supplementary Fig. 7a). The antiferromagnetic-to-paramagnetic transition arises from the GdH₂ hydride[31]. Besides, hydrogenation does not change the magnetic soft nature of the alloy, as indicated in Fig. 4b, which is beneficial for magnetic cooling applications.

From the isothermal magnetization curves (Supplementary Fig. 4c, d), the $\Delta S_M$ of the alloys can be evaluated from the Maxwell relations. As presented in Fig. 4c, the hydronated alloy shows a giant $-\triangle S_M^{pk}$ of 18.7 J kg⁻¹ K⁻¹ under a field change of 5 T, which is comparable to that of the prototype giant-MCE material Gd₅Si₂Ge₂[32,33] and is twice as large as that (9.1 J kg⁻¹ K⁻¹) of the GdCoAl AMA. A rescaled temperature ($\theta = (T - T_p)/(T_r - T_p)$) dependence of the normalized magnetic entropy change $\Delta S_M / \triangle S_M^{PK}$ was analyzed to check the scaling

behavior and illustrate the second-order nature (which is revealed from the Arrott-plot in Supplementary Fig. 4e, f)[14], where $T_p$ is the peak temperature of $\Delta S_M$, and $T_r$ is the reference temperature ($\Delta S_M(T_r)/\triangle S_M^{PK} = 0.7$). As shown in the Supplementary Fig. 4i, all the MCE curves under different fields are collapsed into a single master curve for the GdCoAlH alloy with a dual-phase structure. To highlight the giant-MCE of the hydrogenated alloy, we compare present data with the $-\triangle S_M^{pk}$ values under a field change of 5 T for various Gd-based AMAs and related dual-phase alloys[15,34–39], which are considered the best amorphous magnetocaloric candidates due to the large MCE and free of magnetic hysteresis. As well known, these alloys usually show medium $\Delta S_M$ values due to the second-order magnetic transition nature and thus lack of strong magnetic-structurally coupled transition. Figure 4d indicates that the present hydrogenated alloy breaks through this limitation, which originates from the unique SNDP-GH structure with uniform antiferromagnetic GdH₂ hydride embedded in the amorphous matrix. Moreover, we find that the formation of SNDP-GH structure by hydrogenation is a universal strategy to enhance considerably the $\Delta S_M$ of RE-based AMAs, as seen in several other Gd-, Dy-, and Ho-based AMAs (Supplementary Figs. 5, 6, 8).

## Overview of several property profiles

In practical magnetic cooling application, to manufacture the heat exchanger into required shape and keep the mechanical integrity during temperature and filed variation, the magnetic refrigerants need both high strength and ductility. However, many Giant-MCE materials are subjected to mechanical brittleness and are hard to shape into an effective heat exchanger with the necessary cycling stability. Thus, it is fascinating that the GdCoAlH alloy with SNDP-GH structure shows a

superior combination of mechanical and magnetocaloric properties. To highlight this, we compare the hydrogenated alloy with the typical Gd-based magnetocaloric materials[32,40] (including the Gd metal widely used in prototype magnetic cooling devices and the famous giant-MCE alloy $Gd_5Si_2Ge_2$) in a radar plot containing various mechanical and magnetic properties (Fig. 4e). It is notably that the GdCoAlH alloy possesses an excellent combination of a giant $-\Delta S_M$, ultralow coercivity ($H_c$), high yield strength ($\sigma_y$) and ultimate strength ($\sigma_u$), and large plastic strain ($\varepsilon$), which obviously outperforms other materials. Thus, the SNDP-GH materials have promising applications as magnetic refrigerants with superior magnetocaloric and mechanical performances.

## Discussion

To understand the mechanisms of the superior mechanical and magnetocaloric performances of GdCoAlH, we first briefly illustrate the formation of its unique SNDP-GH structure. It is well-know that RE metals readily form hydrides even by direct action of hydrogen gas at room temperature[41]. Accordingly, during hydrogenation of the GdCoAl powders, the hydrogen prefers to interact with the Gd atoms forming hydride $GdH_2$. When each Gd obtains two hydrogen atoms, hydrogen began to occupy other interstitial sites with different energies gradually approaching the saturation of hydrogen uptake. Due to the excellent glass-forming ability of GdCoAl, the surrounding amorphous matrix is rather stable during hydrogen absorption without any formation of other nanocrystalline phases, even under such a high hydrogen concentration. It was noted that the $GdH_2$ material shows antiferromagnetic transition at around 21 K[31]. Therefore, the antiferromagnetic nature of GdCoAlH at low temperatures originates from the nanosized $GdH_2$ clusters. Since these $GdH_2$ clusters are separated by the amorphous shell, the Néel temperature decreases to around 7 K. The giant $-\Delta S_M$ is related to the uniformly distributed $GdH_2$ clusters with large effective magnetic moment and sharp variation of magnetization with temperature around the $T_N$ of the alloy manifesting as larger $n$ value (Supplementary Fig. 4h, i).

The enhanced strength of the hydrogenated GdCoAlH alloy arises from the dual-phase nanostructure, with the glass phase having higher strength than their crystalline counterparts and the ultrafine nanocrystalline phase having higher strength than their coarse-grained counterparts[42]. The global plasticity of AMAs is determined by the initiation and propagation of SBs generated during compression. The proliferation of SBs leads to the formation of a large plastic zone at the size of decades of micrometers before reaching cavitation. In the GdCoAlH alloy with SNDP-GH structure, the regions containing hydrides are relatively softer than the surrounding matrix due to the expansion of lattice after hydrogenation (Fig. 1i). Since the average distance between the soft regions containing hydrides (average size ~3.6 nm) is much smaller than the plastic zone size (i.e., the critical crack length), thus the localized shears (e.g. dislocations within the NOH) incipiently take place in the NOH[43]. In a CuZr metallic glass composite with B2 CuZr nanocrystals around 2–5 nm dispersed in the amorphous matrix[44], it was found that the B2 CuZr phase deformed by partial dislocations and twinning mechanism. And in a Fe-based BMG with FCC-FeNi local orders of 1–1.5 nm homogenously dispersed in the amorphous matrix, it was reported that FCC-FeNi orders act as both source and sink for the shear transformation zones[45]. Since the size of the shear transformation zone in the GdCoAlH alloy is estimated at around 1.9 nm (Supplementary Fig. 9)[46], which is comparable to the size of NOH and their distance, the deformation of the nanosized hydrides can initiates shear transformation zones in the amorphous shell. On the other hand, the nanosized hydrides can hinder the development and propagation of SBs[45], resulting in the formation of multiple SBs instead of a single SB (Fig. 3e). Hence, the above multiscale deformation mechanism is believed to cause the pronounced plasticity in the GdCoAlH alloy.

In summary, an innovative metal-hydrogen material with a hierarchical SNDP-GH structure is developed. The NOH, with an average size of ~3.6 nm homogenously disperses in the glass matrix with high thermal stability. This unique SNDP-GH structure enhances considerably both the strength and plasticity of the GdCoAlH alloy, which offers a design strategy to overcome the strength-ductility trade-off of AMAs and guide the design of advanced hydrogen embrittlement-resistant alloys. Also strikingly, a giant magnetic entropy change of $18.7\,J\,kg^{-1}\,K^{-1}$ under a field change of 5 T is obtained for the nanostructured hydride-glass composite. This value is 105.5% larger than that of the hydrogen-free sample without change in the second-order transition nature and magnetic softness, which makes GdCoAlH stand out from all Gd-based AMAs. The combination of giant magnetocaloric effect and superior mechanical performance breaks through the traditional thoughts that AMAs have low-to-medium magnetic entropy changes and are intrinsic brittleness, being not suitable for magnetic cooling application. To manufacture related components/devices, sintering such as spark plasma sintering, and ultrasonic forming, etc. could be further used. We also anticipate wide promising applications of the SNDP-GH materials related to their excellent mechanical and other functional properties, in hydrogen separators, hydrogen reformers, and other energetic applications.

## Methods

### Sample preparation and hydrogenation

RE-based alloy ingots with nominal compositions of $Gd_{55}Co_{17.5}Al_{27.5}$, $Gd_{55}Ni_{17.5}Al_{27.5}$, $Dy_{55}Co_{20}Al_{24}Si_1$, and $Ho_{55}Co_{20}Al_{24}Si_1$ were prepared by arc-melting a mixture of pure elements in a titanium-gettered argon atmosphere. The purities of Gd, Tb, Dy, and Ho elements are better than 99.9 wt.%, and those of Co, Ni, Al, and Si elements are better than 99.99 wt.%. The amorphous powders were prepared by the gas atomization method. The powders with the size of 25–30 μm were selected for the subsequent tests. The isothermal hydrogenation experiment was performed on a Sieverts-type apparatus (Advanced Materials Corporation, No. 0360Q). The amorphous powders were encapsulated in a sample chamber with a vacuum atmosphere. Before hydrogenation, the amorphous powders were heated from room temperature to 473 K at a heating rate of 10 K/min and isothermally annealed for 30 min for activation. Then, the hydrogen pressure was set at 5 MPa, and isothermal hydrogenation was performed at 473 K until saturation. The sample packaging and collection were completed in a glove box.

### Analytical methods

The thermal properties of amorphous powders were analyzed using differential scanning calorimetry (DSC, Netzsch DSC 404 F3) at a heating rate of 20 K/min, and the onsets of glass transition temperature ($T_g$) and crystallization temperature ($T_x$) were identified by the tangent method. The structures of powders were examined by X-ray diffraction (XRD, Bruker D8 Discover) using Cu-Kα radiation. The morphology was characterized by scanning electron microscopy (SEM, Thermo Fisher Verios 5UC). Furthermore, the structural features at the nanoscale were investigated by a spherical aberration-corrected transmission electron microscopy (TEM, Thermo Fisher Spectra 300). A focused ion beam (FIB, Thermo Fisher Helios 5UX) was used to prepare the TEM samples. Further energy-dispersive X-ray spectroscopy (EDS) analysis was conducted using Thermo Fisher Scientific's Super-X windowless EDS detector at an acceleration voltage of 300 kV. Electron energy loss spectroscopy (EELS) was used to analyze the hydrogen in the sample.

### Mechanical response measurements

Micropillars of the RE-based powders were fabricated with a FIB. The aspect ratio (height/diameter) of each micropillar was set as 2 with a

diameter of 500 nm, 3 µm, and 5 µm. Micro-compression tests were conducted at room temperature under displacement-control mode and at a strain rate around $5 \times 10^{-3} \, s^{-1}$, using the nanoindenter's high-load mode with a flat punch diamond tip (Bruker TI980). Each test was performed on five pillars. To estimate the size of the shear transformation zone, nanoindentation creep deformation tests were performed using a NanoTest Vantage (Micro Materials Ltd) with a standard Berkovich diamond indenter under load control model. In this measurement, the samples were loaded to 10 mN with constant loading rates of 0.1, 1, 5, and 10 mN/s; and then held 100 s at the load limit; finally unloaded at the same rate. For each sample and loading rate, five effective indentation tests were performed.

### Magnetocaloric response measurements

Magnetic properties of the powders were performed by a SQUID magnetometer (MPMS, Quantum Design) and a physical properties measurement system (PPMS6000, Quantum Design). To be specific, temperature dependences of the DC magnetization from 2 to 200 K under an applied magnetic field after cooling the sample to 2 K under the field (FC) of 0.01 T or zero field (ZFC), respectively, were measured using an MPMS. Isothermal magnetization curves were recorded from 0 to 5 T with a slow sweeping rate of the field at different temperatures from 2 to 180 K. The AC magnetic measurements with a probing field of 4 Oe under different frequencies were carried out in a PPMS.

### Data availability

All data needed to evaluate the conclusions in the paper are present in the paper and/or the supplementary information. Data were also available from the corresponding author upon request.

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

## Acknowledgements

This work was supported by the National Natural Science Foundation of China (Grant Nos. 52231005, 52301212, 51971061, 52101193, NSFC-NSAF, and U2330111), Natural Science Foundation of Jiangsu Province, China (BK20221473), and Guangdong Major Project of Basic and Applied Basic Research, China (Grant No. 2019B030302010).

## Author contributions

L.S., Q.L., and B.S. designed the research project; L.S., M.Z., L.X., J.C., and Q.Y. fabricated the samples and performed the magnetic and mechanical experiments and structural tests; Y.Z. performed the hydrogenation experiments; Q.L., H.K., and B.S. conceptualized and supervised the research; Q.L. and L.S. wrote the draft of the manuscript; W.W. contributed to the general discussion of the results. All authors commented on the manuscript.

## Competing interests

The authors declare no competing interests.
