## [Peer Review File · Nature Communications]

Dual-phase nano-glass-hydrides overcome the strength-ductility trade-off and magnetocaloric bottlenecks of rare earth based amorphous alloysREVIEWER COMMENTS

Reviewer #1 (Remarks to the Author):

Review Report

Dual-phase nano-glass-hydrides overcome the strength-ductility trade-off and magnetocaloric bottlenecks of rare earth based amorphous alloys
by Liliang Shao et al.

This paper presents a comprehensive investigation into the formation of nano-hydrides, GdH₂, within the amorphous glass matrix, showcasing noteworthy functional properties. These properties include hydrogen embrittlement, leading to a hydrogen-induced brittle-to-ductile transition, improvements in mechanical properties such as enhanced strength and strain deformation, and notable magnetocaloric effects. The findings presented in this study warrant publication in Nature Communications. The following recommendations are provided:

Kindly rephrase lines 145 and 146 for clarity and precision.

The hydrogenation process occurred at 473 K under a pressure of 5 MPa. Was any activation procedure employed before the hydrogenation? If affirmative, please ensure to incorporate the details in the manuscript.

Reviewer #2 (Remarks to the Author):

Dear authors,

This manuscript demonstrated excellent mechanical and magnetic properties of hydrogenated GdCoAl alloys. Although interesting information is contained, there are some questions before accepting your manuscript. My questions are provided as follows:

Q1) This investigation is done by using GdCoAl powders prepared by high-pressure argon gas atomization. The reviewer accepts that the powders had the excellent mechanical and magnetic properties after hydrogen charging; however, I do not still understand how real components are manufactured from the powders and those components possess similar excellent properties to the powders. At least, if we assume components in high-pressure hydrogen gas, we cannot use GdCoAl in the form of powders. In a such situation, are the real components manufactured by sintering and additive-manufacturing?

Q2) The reviewers should mention how the powders GdCoAl were hydrogenated.

Responses to the Reviewers' comments

ID: NCOMMS-23-61120-T

Title: Dual-phase nano-glass-hydrides overcome the strength-ductility trade-off and magnetocaloric bottlenecks of rare earth based amorphous alloys

Thanks for the reviewers' comments concerning our manuscript. Those comments are valuable and helpful for revising and improving our paper, as well as the important guiding significance to our research. We have studied the comments carefully and have made correction which we hope meet with approval. We have denoted our updates in the revised manuscript using blue text for easy identification.

Reviewer #1:

Comments to the Author(s)

Dual-phase nano-glass-hydrides overcome the strength-ductility trade-off and magnetocaloric bottlenecks of rare earth based amorphous alloys by Liliang Shao et al.

This paper presents a comprehensive investigation into the formation of nano-hydrides, GdH_2 , within the amorphous glass matrix, showcasing noteworthy functional properties. These properties include hydrogen embrittlement, leading to a hydrogen-induced brittle-to-ductile transition, improvements in mechanical properties such as enhanced strength and strain deformation, and notable magnetocaloric effects. The findings presented in this study warrant publication in Nature Communications. The following recommendations are provided:

Comment 1. Kindly rephrase lines 145 and 146 for clarity and precision.

Author reply: Thank the reviewer very much for the professional suggestion. The sentence "The amorphous matrix forms a shell with thickness of 1-4 nanometers to wrap up the NOH." has been rephrased in the revised manuscript. For your convenience, they are shown below: "The amorphous matrix serves as a shell with thickness of 1-4 nm, which wraps up the NOH structure." Line 13, Page 6.

In addition, we also carefully checked the sentences and grammars of the entire text.

Comment 2. The hydrogenation process occurred at 473 K under a pressure of 5 MPa. Was any activation procedure employed before the hydrogenation? If affirmative, please ensure to incorporate the details in the manuscript.

Author reply: Thank the reviewer very much for the insightful question. An isothermal annealing was employed before the hydrogenation. The procedure was added in Methods of the revised manuscript. For your convenience, they are shown below: “The amorphous powders were encapsulated in a sample tube with vacuum atmosphere. Before hydrogenation, the amorphous powders were heated from room temperature to 473 K at a heating rate of 10 K/min and isothermally annealed for 30 min for activation. Then, the hydrogen pressure was set as 5 MPa, and isothermal hydrogenation was performed until saturation. The sample packaging and collection were completed in a glove box.” Line 8, Page 12.

Reviewer #2

Comments to the Author(s)

This manuscript demonstrated excellent mechanical and magnetic properties of hydrogenated GdCoAl alloys. Although interesting information is contained, there are some questions before accepting your manuscript. My questions are provided as follows:

Comment 1. This investigation is done by using GdCoAl powders prepared by high-pressure argon gas atomization. The reviewer accepts that the powders had the excellent mechanical and magnetic properties after hydrogen charging; however, I do not still understand how real components are manufactured from the powders and those components process similar excellent properties to the powders. At least, if we assume components in high-pressure hydrogen gas, we cannot use GdCoAl in the form of powders. In a such situation, are the real components manufactured by sintering and additive-manufacturing?

Author reply: We would like to thank the reviewer very much for these constructive comments. In this study, the characterizations of mechanical and magnetic properties, and structural

analyses were all performed using powder material obtained by gas atomization, which was stated in the Methods section. This study focuses on the structure, mechanical and magnetocaloric effect of the GdCoAlH sample, and does not involve the manufacture of components.

The reason that amorphous powders were used in this study is that powder materials possess large specific surface area, which is beneficial to hydrogenation. By all means, the reviewer proposed a valuable idea, i.e., manufacturing bulk components on base of amorphous powders. The fabrication of bulk component can be achieved through sintering such as spark plasma sintering [*Nature* **626** (2024) 779-784], and ultrasonic forming [*Nature Communications* **14** (2023) 6305]. Maybe, additive-manufacturing is also feasible, if the temperature is confined within the dehydrating temperature (~ 600 K) of the GdCoAlH sample, and thus the excellent properties of powder will inherit to bulk component. In the future, we may try to manufacture bulk components based on the hydrogenated powders, and we hope more researchers focus on this topic to promote the application of the magnetocaloric material.

In the revised manuscript, we have added some perspectives according to your comments. For your convenience, they are shown below: “To manufacture related components/devices, sintering such as spark plasma sintering, and ultrasonic forming, etc. could be further used.”
Line 19, Page 11.

Comment 2. The authors should mention how the GdCoAl powders were hydrogenated.

Author reply: Thanks for kindly reminding us to clarify this point. The detailed procedure of hydrogenation was added in Methods of the revised manuscript. For your convenience, they are shown below: “The isothermal hydrogenation experiment was performed on a Sieverts-type apparatus (Advanced Materials Corporation, No. 0360Q) [*Advanced Materials* **35**(2023) 2303173]. The amorphous powders were encapsulated in a sample tube with vacuum atmosphere. Before hydrogenation, the amorphous powders were heated from room temperature to 473 K at a heating rate of 10 K/min and isothermally annealed for 30 min for activation. Then, the hydrogen pressure was set as 5 MPa, and isothermal hydrogenation was performed at 473 K until saturation. The sample packaging and collection were completed in a

glove box.” Line 7, Page 12.

In addition, the format was revised according to the *Nature Communications* formatting instructions.

REVIEWERS' COMMENTS

Reviewer #1 (Remarks to the Author):

The paper was carefully revised. All points raised in my first review were answered adequately. The paper now deserves to be published in NCOMMS.

Reviewer #2 (Remarks to the Author):

Dear authors,

Thank you for your revisions.
I satisfied your responses and do not have any comments.